# The Preoperative Prognosticators of Surgical Margins (R0 vs. R1) in Pelvic Exenteration—A 14-Year Retrospective Study from a Tertiary Referral Centre

**DOI:** 10.3390/cancers17223679

**Published:** 2025-11-17

**Authors:** Sabina Ioana Nistor, Roman Mykula, Richard Bell, William Gietzmann, Mahmoud Awaly, Alaa Elzarka, Jennifer Thorne, Jacopo Conforti, Federico Ferrari, Nicholas Symons, Hooman Soleymani majd

**Affiliations:** 1Department of Gynaecological Oncology, Churchill Hospital, Oxford University Hospitals, National Health Service (NHS) Foundation Trust, Oxford OX3 7LE, UK; sabina.nistor@ouh.nhs.uk (S.I.N.); mahmoud.awaly1@nhs.net (M.A.); a_elzarka09@alexmed.edu.eg (A.E.);; 2Department of Plastic Surgery, Oxford University Hospitals, National Health Service (NHS) Foundation Trust, Oxford OX3 7LE, UK; roman.mykula@ouh.nhs.uk; 3Department of Urology, Oxford University Hospitals, National Health Service (NHS) Foundation Trust, Oxford OX3 7LE, UK; richard.bell@ouh.nhs.uk (R.B.); william.gietzmann@ouh.nhs.uk (W.G.); 4Department of Obstetrics and Gynaecology, John Radcliffe Hospital, Oxford University Hospitals, National Health Service (NHS) Foundation Trust, Oxford OX3 9DU, UK; jennifer.thorne@nhs.net; 5Department of Clinical and Experimental Sciences, University of Brescia, 25123 Brescia, Italy; federico.ferrari@unibs.it; 6Department of Colorectal Surgery, Oxford University Hospitals, National Health Service (NHS) Foundation Trust, Oxford OX3 9DU, UK; nicholas.symons@ouh.nhs.uk; 7Nuffield Department of Women’s and Reproductive Health, University of Oxford, Oxford OX3 9DU, UK

**Keywords:** pelvic exenteration, post-radiotherapy, predicting factors, R0 resection

## Abstract

Pelvic exenteration is one of the most extensive operations performed for gynaecological cancers that return or persist after radiotherapy. It involves removing the cancer along with nearby pelvic organs, followed by reconstructive surgery. The main aim is to achieve “R0 resection,” meaning all visible and microscopic cancer is removed, which offers the best chance of long-term survival. We looked at 27 women who had surgery at Oxford University Hospitals between 2011 and 2024, after radiotherapy. Their cancers started in the cervix, uterus, vagina, or vulva. The average age was 63. Surgeons achieved complete removal in almost 80% of cases. Complications happened during surgery in about one in three women, and serious post-surgery problems in about one in five. Younger patients, with smaller tumour size, and those with cervical cancer and reduced blood loss during surgery were more likely to achieve full removal.

## 1. Introduction

Pelvic exenteration (PEx) is a complex surgery considered for locally advanced or recurrent pelvic malignancies, entailing a radical en-block resection of multiple adjacent pelvic organs, followed by reconstructive surgery.

It was first described by Alexander Brunschwig in 1948 as a palliative procedure for recurrent cervical cancer [1]. Mortality rates of PEx have reduced from 23% in Brunschwig’s series to 1.7% in modern series [2], while the reported 5-year overall survival range is 32–70% for gynaecological malignancies [3,4,5], making these procedures “curative” and acceptable for well-selected patients.

Modifications of the surgical technique have extended the en-block resection to include, in addition to pelvic organs, adjacent structures such as pelvic sidewall muscles, nerves and major vessels [6,7,8], or bony structures (composite exenterations) [9,10,11], in an attempt to obtain complete resection with negative margins (achieve R0).

Compelling evidence supports the importance of achieving R0 resection, the Achilles heel of pelvic exenteration, which is associated with improved disease-free survival [12,13,14,15,16,17] and overall survival [4,12,14,15,16,17,18].

A retrospective cohort including 1293 patients from 22 centres undergoing PEx for non-colorectal malignancies, 523 of them gynaecological, demonstrated that R0 resection was the main factor associated with long-term survival in multivariate analysis. The 3-year overall survival rates for patients having R0 resection for advanced or recurrent endometrial, ovarian, cervical, or vaginal malignancy were 48%, 40.6%, 49.4%, and 43.8%, respectively, while R1 and R2 resections had significantly lower overall survival: 22.2%, 30.3%, 37%, and 12.5%, respectively [2].

Palliative exenteration remains a controversial practice, given the high rates of associated morbidity in patients with a limited overall survival [19], as well as a doubtful palliation benefit [14]. Quyn et al. report that, following palliative PEx, patients experienced a sustained decline in QoL after surgery until death [20]. Therefore, all efforts should be made in the preoperative period to select those patients who are likely to have a true survival benefit following this marathon surgery [21].

The objective of our study was to identify factors influencing R0 resection in an irradiated field, including patient and tumour characteristics or surgical aspects, in order to facilitate optimal patient selection and define a surgical roadmap for performing exenterative surgery.

## 2. Materials and Methods

Our retrospective observational cohort included consecutive patients who underwent pelvic exenteration for gynaecological malignancies at Oxford University Hospitals between 1 January 2011 and 31 December 2024. Oxford is the tertiary referral centre for the Thames Valley Cancer Alliance Network, which includes five recruiting sites and serves a catchment area of 2.3 million.

The primary outcome was margin status. Secondary outcomes were intraoperative and postoperative complications.

Data was collected from paper and electronic patient records. This service evaluation protocol was registered in accordance with the Oxford University Hospitals Trust requirements (registration number 8912). The study was conducted in accordance with the Declaration of Helsinki (as revised in 2013). The data collected were anonymised. Informed consent was obtained from all patients as part of standard clinical practice to allow for data collection and analysis for research purposes. No remuneration was offered to the patients enrolled in this study.

### Definitions

We have defined PEx as follows:

An anterior exenteration involved en-block excision of the gynaecological organs and the urinary bladder. A posterior exenteration included the en-block excision of the gynaecological organs and the rectosigmoid with or without the anus and anal sphincters. A total exenteration comprised of the en-block excision of the urinary bladder, gynaecological organs, and recto-sigmoid ± anus.

We have used the cranio-caudal subclassification of PEx proposed by Magrina et al. (1990) [22] to facilitate a clear understanding of the extent of resection and anatomical changes associated with each type of surgery. A type I PEx is supralevator. A type II PEx is infralevator, with the dissection extending inferiorly to include the levator ani muscles and urogenital diaphragm. A type III PEx includes vulvectomy ± excision of perineal tissues [22].

With regards to margin status, we have defined R0 resection as clear histopathological margins, with tumour > 1 mm from margin, while R1 resection entailed the presence of microscopic residual disease < 1 mm from margin. R2 resection was defined as the presence of macroscopic residual disease at the time of surgery.

A number of patient characteristics were collected from patient notes. Demographic data included age and body mass index (BMI). Medical comorbidities were assessed using the Karnofski Performance Status (PS), the American Society of Anaesthesiologists (ASA) Physical Status classification system, and the Charlson Comorbidity Index (CCI).

All patients followed the Oxford pathway for pelvic exenterations/laterally extended endopelvic resections (LEER) (Figure 1).

Prior to surgery, the Oxford checklist was completed to ensure all relevant preoperative steps have been undertaken (Table 1).

The following surgical details were collected from the operation notes: type of exenteration (anterior/posterior/total and type I/II/III), estimated blood loss (ELB), whether laterally extended endopelvic resection (LEER) was performed, type of urinary diversion, bowel surgery and plastic surgery reconstruction, and intraoperative complications.

From the histopathology reports, we collected the histology diagnosis (squamous cell carcinoma, adenocarcinoma, other), the tumour size, and the margin status (R0/R1/R2).

Postoperative complications within 30 days from the date of surgery were defined using the Clavien–Dindo classification. Data regarding postoperative complications and length of stay (LOS) was collected from patient records.

Statistical analysis was performed using SPSS v. 30.0. Baseline demographic and clinical characteristics were summarised by continuous and categorical variables. Categorical variables were compared with the Chi-square test. When the assumptions required for Chi-square test were not met, Fisher’s exact test was employed. Differences between continuous variables were analysed with the Independent Student’s *t*-test. Ordinal variables were analysed using the Kruskall–Wallis non-parametric test. A value of *p* < 0.05 was considered statistically significant. Univariable and multivariable logistic regression was performed using margin status as the dependent variable.

## 3. Results

We have identified a total of 267 cases. We have excluded 24 patients who either had incomplete records or were not found to have had true exenterations after review of the notes. Out of the remaining 243 cases, we excluded 206 patients for which en-block resection performed during ovarian cytoreductive surgery was coded as “posterior exenteration”—this coding captured only a portion of the cytoreductive surgeries for advanced ovarian cancer performed in our centre. Thirty-seven patients underwent pelvic exenteration for vulval, vaginal, cervical, or uterine malignancies. From these 37 patients, we have excluded 10 cases who did not receive radiotherapy (RT) prior to the pelvic exenteration (Figure 2).

We identified 27 patients with previous radiotherapy treatment, who had a pelvic exenteration for non-ovarian gynaecological malignancies. We only included patients who underwent PEx after RT, as performing major surgery in a previously irradiated field presents additional critical surgical challenges.

### 3.1. Patient and Tumour Characteristics

The median age was 63 years (range, 41–81) and median body mass index (BMI) was 27 (range, 17–45). Most patients had a performance status (PS) of 0 or 1, as follows: 18 (66.7%) had a PS of 0, 8 (29.6%) had a PS of 1, and 2 (3.7%) had a PS of 2. The American Society of Anaesthesiologists (ASA) score was 1 for 18 patients (66.7%), 2 for 7 patients (27.9%), and 3 for 2 patients (7.4%). The median Charlson Comorbidity Index was 2 (range 0–5) (Table 2).

With regards to the primary tumour site, 3 (11.1%) patients had vulvar cancer, 4 (14.8%) had vaginal cancer, 11 (40.7%) had cervical cancer, and 9 (33.3%) patients had uterine cancer.

The histological diagnosis was squamous cell carcinoma (SCC) for 15 (55.5%) patients, adenocarcinoma (ADK) for 9 (33.3%) patients, carcinosarcoma for 2 (7.4%) patients, and serous carcinoma for one (3.7%) patient.

The majority of our patients (14, 51.9%) had International Federation of Gynaecology and Obstetrics (FIGO) stage 1 disease at diagnosis, 6 (22.2%) had stage 2 disease, 4 (14.8%) had stage 3, and 3 (11.1%) had stage 4 disease.

The primary treatment type was chemoradiotherapy only for 12 (44.4%) patients, surgery and radiotherapy for 8 (29.6%) patients, and all three treatment types (chemotherapy, radiotherapy, and surgery) for 7 (25.9%) patients.

At the time of exenteration, 18 (66.7%) patients had recurrent disease, 6 (22.2%) patients had persistent disease, and 3 (11.1%) patients had a new type of malignancy.

The median time to recurrence was 33 months (range, 8–348). The median pre-operative tumour size was 4.1 cm (range, 0.9–6.7 cm) (Table 2).

### 3.2. Surgical Technique and Morbidity

The median estimated blood loss (EBL) for our patients was 800 mL (range, 150–2500), and the median LOS was 23 days (range, 5–116).

Seventeen (63%) patients had a total exenteration, eight (29.6%) patients had a posterior exenteration, and two (7.4%) patients had an anterior exenteration. Using the classification proposed by Magrina et al. (1990) [22], 11 (40.7%) patients had a type III, 1 (3.7%) patient had a type II, and 15 (55.6%) had a type I exenteration. Four (14.8%) patients had LEER.

A urinary diversion was required in 19 (70.4%) patients, following cystectomy. All urinary diversions were incontinent conduits. Ten (37%) patients had an ileal conduit, seven (25.9%) of which had a Wallace ileal conduit and three (11.1%) had a Bricker ileal conduit. Nine (33.3%) patients had a colonic conduit; two (7.4%) had a Wallace colonic conduit, and (25.9%) had a Bricker colonic conduit.

Twenty-five (92.6%) of patients underwent colorectal surgery. The most common colorectal procedure performed was Hartmann’s procedure with end colostomy 16 (59.3%). Three patients (11.1%) required abdomino-perineal resection (APR) and end colostomy. Six (22.2%) patients underwent other colorectal procedures.

Nine (33.3%) patients required plastic surgery, and the most common procedure was a vertical rectus abdominis myocutaneous (VRAM) flap, performed for 6 (22.2%) patients. Two (7.4%) patients had and antero-lateral thigh (ALT) flap. One (3.7%) patient had bilateral gracilis flaps, following intraoperative VRAM flap failure due to compromised blood supply. (Table 3)

Eight patients (29.6%) experienced intraoperative complications. Three of them sustained an iliac vein injury, with documented EBLs of 1.5 L, 2 L, and 2.5 L, respectively. Two patients had cystotomies, which were identified and repaired intraoperatively with no postoperative issues. Two patients had enterotomies, one of them requiring small bowel resection and primary anastomosis. One patient had an ischaemic VRAM flap, due to ligation of the inferior epigastric artery during the surgical excision. Perineal reconstruction was performed using bilateral gracilis flaps instead. Finally, one patient had an ischaemic Wallace ileal conduit, which was removed and replaced with a Wallace sigmoid conduit.

Six (22.2%) of our patients had significant postoperative complications (Clavien–Dindo grade III or IV), in the first 30 days post-surgery. Two patients had Clavien–Dindo grade IIIA complications: one developed a para-colic collection, which was drained under ultrasound-guidance, and the second one required a nephrostomy on a single kidney.

Four (14.8%) patients had Clavien–Dindo grade IIIB complications. One of them was found to have a congested VRAM flap on day 2 post surgery. She was taken back to theatre and 50% of the flap was thought to be compromised. The compromised tissue was debrided, and the perineum was reconstructed using a left gracilis flap and a right superior gluteal artery perforator (SGAP) rotational flap and Z-plasty. She later developed a perineal wound dehiscence, which was managed conservatively with a Vacuum-Assisted Closure (VAC) dressing and antibiotics. She also received treatment for urosepsis during admission.

A second patient developed sepsis and a deterioration in kidney function on day 9 post surgery. Computerised tomography (CT) was performed, which identified an entero-vaginal fistula and a fistula between the ileal conduit and the small bowel. She was taken to theatre on day 12 post surgery, with the intention to repair the fistulae. On entry to the abdomen, the small bowel entered a pelvic inflammatory phlegmon which was not thought safe to remove due to risk of multiple enterotomies. A defunctioning proximal loop ileostomy was created. The urine was diverted using bilateral nephrostomies. She was discharged home on day 20.

Another patient developed an ischaemic right leg on day 1 post surgery, and she returned to theatre for embolectomy of an external iliac artery and superficial femoral artery thrombus. The embolectomy unfortunately failed and the patient experienced recurrent ischaemia secondary to compartment syndrome. She was transferred to theatre a second time for a femoral–femoral cross-over graft and right lower leg fasciotomy. She also experienced a pelvic collection, which was managed conservatively.

Finally, a fourth patient developed dehiscence of the vaginal and anal stumps and a recto-vaginal fistula, which was managed conservatively using endo-sponge therapy (Table 4).

### 3.3. Margin Status

Twenty-one (77.8%) of our patients had resection of all microscopic and macroscopic disease (R0 was achieved); four (14.8%) had no residual tumour, while seventeen (63%) had residual carcinoma in the specimen, with microscopically-clear margins.

Of the six (22.2%) patients with microscopically-involved margins, four (14.8%) had one margin involved, one (3.7%) had two margins involved, and one (3.7%) had three margins involved. None of the patients had macroscopic residual disease (R2) (Table 5).

### 3.4. Lymph Node Status

With regards to lymph node status, none of our patients had macroscopically-involved lymph nodes. Microscopic lymph node involvement was identified in two cases. The first case, a 74-year-old patient undergoing pelvic exenteration for uterine carcino-sarcoma, with a history of anal squamous cell carcinoma treated with chemoradiotherapy, was found to have microscopic involvement of para-aortic lymph nodes by metastatic anal squamous cell carcinoma. This patient had clear margins (R0). The second patient, a 63-year-old undergoing posterior exenteration for recurrent vulval squamous cell carcinoma, was found to have a microscopically involved left femoral lymph node. This patient had a tumour measuring 8.9 cm on histology, and the margins were microscopically involved (R1). We did not include the lymph node status in our statistical analysis as we only had two cases with lymph node involvement, one by a different tumour.

### 3.5. Statistical Analysis

Univariable analysis was performed to identify associations between a number of variables and R0 status (Table 6).

Our paper found a statistically significant association between the patient’s age and R0 status (*p* value 0.035, Independent *t*-test), with a median age of 61 in the R0 group versus 70 in the R1 group.

There was a significant association between tumour size and R0 status. This applied to both estimated pre-operative tumour size based on imaging (*p* value 0.014 Independent *t*-test) and actual tumour size on the histopathology specimen (*p* value 0.006 Independent *t*-test). The mean tumour size was 3.4 cm in the R0 group, both on pre-operative and on histopathological assessment. In the R1 group, the mean tumour size was 5 cm on imaging and 6 cm on histopathological examination.

The mean estimated blood loss was 653 mL in the R0 group, versus 1267 mL in the R1 group, with a *p* value of 0.06, just above significance level (Independent *t*-test).

Univariable logistic regression was performed to assess the effect of multiple factors on the likelihood of achieving R0, and the results are presented in Table 7.

Tumour size on imaging, tumour size on histology, and EBL were found to be statistically significant in predicting the likelihood of achieving R0.

Multivariable logistic regression was performed, using margins status as dependent variable and age, EBL and tumour size on histology as independent variables (Table 8). None of these three variables remained significant, likely due to limited number of events (6 patients with R1) and a small sample size with only 27 patients.

## 4. Discussion

Multiple studies have demonstrated a negative association between larger tumour size on preoperative assessment and both progression-free and overall survival following exenterative surgery [2,18,23,24], and various tumour size thresholds have been suggested as criteria for offering PEx: 3 cm [4,25,26], 4 cm [16,17], and 5 cm [27].

This association is likely a result of lower R0 resection rates in patients with larger tumours.

Our study identified a significant association between tumour size and R0 resection, both on univariable analysis (Independent *t*-test) and on univariable logistic regression. The mean tumour size on histopathological examination was 3.4 cm in the R0 group vs. 6 cm in the R1 group.

These results are in keeping with existing literature. In a retrospective study including 151 patients with gynaecological, colorectal, and urological malignancies undergoing pelvic exenteration, Smith et al. (2017) [28] found that there was an 11% increase in the risk of positive margins for every 1 cm increase in size of tumour (OR 1.11; 95%CI 1.02–1.22). They also found that tumours > 4 cm in size were more likely to recur (65%) compared to tumours ≤ 4 cm (42%) and cases with no residual tumour on final pathology (20%), *p* = 0.016 [28].

One of the most daunting tasks for the multidisciplinary team pre- and intraoperatively is to distinguish between the tumour tissue and radiation-induced fibrosis or local inflammation [29]. Pre-operatively, an MRI of the pelvis can assess the relationship between the tumour and various adjacent anatomical structures such as ureters, nerves, and iliac vessels [30], while Diffusion Weighted Imaging sequences add functional information and can help distinguish fibrosis from tumour [31]. PET CTs, generally used for detecting lymph node metastases and extra-pelvic disease [31], also aid in defining tumour extent and differentiating fibrosis from metabolically active malignant tissue [32]. However, preoperative tumour size assessment remains a challenge in the post-radiotherapy field. We found a mean difference of 1.4 cm (range 0–4.9 cm) between tumour size on imaging and tumour size on histology. For this reason, while there is no general consensus globally, we recommend obtaining a generous margin of 2–3 cm of healthy tissue at the time of exenteration, when anatomically possible, keeping in mind that, following paraffin fixation, there is linear tissue shrinking by approx. 15% and a 30–40% loss in tissue volume.

Our paper found a statistically significant association between the patient’s age and R0 status (*p* value 0.035, Independent *t*-test), with a median age of 61 in the R0 group versus 70 in the R1 group. We believe this association may be justified by the presence of more advanced fibrosis in older patients, making intraoperative assessment of margins problematic. While advancements in technology have enabled clinical oncologists to target tissue with higher precision, collateral damage to normal tissue remains an unavoidable consequence of radiotherapy. Chronic inflammatory and fibrotic changes typically occur up to one year after completing treatment and worsen over time. Changes include progressive tissue induration, rigid stromal encasement of capillaries and sinusoids that become distorted and dilated, entrapment of neurological structures, dryness, and atrophy [33]. Trauma or infection may lead to further vascular compromise and result in ulceration or necrosis [33]. Furthermore, older age at time of radiotherapy has been identified as a strong independent risk factor for developing radiation-induced toxicity [34], likely due to a reduced ability to repair damaged tissues at a more advanced age. Comorbidities affecting the microcirculation such as diabetes, cardiovascular disease, and immobility may further accelerate fibrotic progression. We found that intraoperative EBL was statistically significant in predicting the likelihood of achieving R0, with a mean EBL of 653 mL in the R0 group, versus 1267 mL in the R1 group. This is not surprising, given the deleterious effect of bleeding on the surgical field, reducing visibility and making tissue planes demarcation more challenging. Bleeding also increases the pace of surgery and heightens the stress levels of the surgical team, therefore making surgical errors more likely. It is our routine practice to start any PEx procedure by opening the pelvic sidewalls, exposing the pararectal, para-vesical and Latzko spaces and achieving devascularisation of the pelvis by ligating the anterior branch of the internal iliac artery 3 cm distal to the common iliac artery bifurcation. Higher blood loss could also represent a surrogate marker for more advanced disease increasing the difficulty of surgery.

We identified a statistically significant association between primary tumour site and R0 resection (*p* = 0.036). While all cervical cases achieved negative margins, only 50% of non-cervical tumour types did.

The observed differences likely reflect the distinct biological behaviours and treatment sensitivities among gynaecological malignancies. Cervical cancer is generally both radio- and chemosensitive, particularly in squamous histology, and tends to recur centrally, making it more amenable to complete surgical clearance. In contrast, endometrial cancers are often less radiosensitive and more prone to distant dissemination, reducing the likelihood of durable local control. Vulval cancers, while predominantly squamous, comprise two biologically distinct subtypes: HPV-associated (p16-positive) tumours, which are typically more radiosensitive and occur in younger patients, and p53-mutated (HPV-independent) tumours, which are more common in older women, often arise in the setting of chronic dermatoses, and are less responsive to radiotherapy. Vaginal cancers, though rare and often managed similarly to cervical cancer, may exhibit variable responses to radiation due to anatomical proximity to radiosensitive organs and reduced vascular perfusion in some segments of the vaginal wall, both of which limit the radiation dose that can be safely delivered and affect tissue tolerance. These differences underscore the importance of tailoring exenterative surgery to tumour biology, histological subtype, and recurrence pattern when selecting candidates for this complex and highly morbid procedure and support pelvic exenteration as being most effective in the management of centrally recurrent cervical cancer.

Our study is the first specifically looking at R0 and complication rates in previously irradiated patients undergoing a pelvic exenteration for a non-ovarian gynaecological malignancy. In Table 9, we have summarised the findings of 14 retrospective studies on pelvic exenteration published in the last 10 years (2015–2025), demonstrating data gaps, particularly in relation to reporting R0 rates—the main factor associated with survival—and post-radiotherapy status. Our study compares favourably with existing literature in terms of an R0 resection rate of 77.8% (54.2–81.6%) and a 30-days severe complications rate (Clavien–Dindo grade 3 and 4) of 23.2% (21.2–74%).

This study has several limitations that should be acknowledged. Firstly, its retrospective observational design is inherently subject to selection and information bias, as data were collected from existing medical records with variable completeness and accuracy. The small sample size (n = 27) limited the statistical power of the analyses and likely contributed to the fact that, although multivariable analysis was performed, none of the variables retained statistical significance. This limits the ability to draw firm conclusions regarding independent predictors of R0 resection. Pelvic exenteration is a rare procedure, reserved for a very limited number of patients, which inevitably results in a paucity of data. We believe it was important to restrict inclusion criteria to post-radiotherapy cases only, as achieving R0 resection in a previously irradiated field poses specific challenges. In gynaecological malignancies, radiotherapy typically delivers higher biologically effective doses, often exceeding 80 Gy EQD2 when combining external beam and brachytherapy to central pelvic tissues, resulting in extensive fibrosis and obliteration of normal tissue planes. In contrast, colorectal and urological cancers usually receive lower or more conformally delivered doses—approximately 45–50 Gy for rectal cancer and 64–80 Gy for bladder cancer—targeting smaller volumes and sparing much of the surrounding soft tissue. Consequently, surgery after gynaecological radiotherapy is uniquely demanding, with increased risks of vascular and ureteric injury and difficulty distinguishing tumour from post-treatment fibrosis. Furthermore, we excluded ovarian cases, as en-bloc resections performed during ovarian cytoreductive surgery are typically supralevator and undertaken in non-irradiated fields, representing a distinct disease biology, treatment paradigm, and survival outlook. We also excluded pelvic exenterations performed for non-gynaecological malignancies, and patients who underwent LEER only without PEx. Our sample size is comparable to other single-centre publications from major referral centres when accounting for these stricter inclusion criteria (Table 9). We believe that a future prospective, multicentre collaboration would be highly valuable to validate and build upon our findings. Positive pelvic, para-aortic [44], and inguino-femoral [45] nodal involvement is known to be a key predictor of reduced overall survival. Our study only identified one patient with an unexpected microscopic involvement of para-aortic lymph nodes by a different tumour (anal squamous cell carcinoma), and one patient with microscopic involvement of a femoral lymph node; therefore, we were not able to include lymph node status in the statistical analysis. Finally, we do not present long-term oncological data in this study, but we will endeavour to report this once our patient cohort has completed follow-up. 

## 5. Conclusions

From a clinical standpoint, our findings emphasise the importance of meticulous preoperative evaluation, multidisciplinary planning, and judicious patient selection. The patients most likely to benefit from pelvic exenteration in a post-radiotherapy field are younger individuals with smaller, centrally recurrent tumours—particularly of cervical origin—where complete (R0) resection is achievable. Intraoperatively, careful control of blood loss, achieved through early devascularisation of the pelvis, is essential to optimising surgical precision and outcomes.

In summary, preoperative tumour size, primary tumour site, and patient age should be key considerations in selecting suitable candidates for pelvic exenteration following radiotherapy, while minimising intraoperative blood loss remains critical to maximising the likelihood of achieving R0 resection—the single most important determinant of survival in this complex procedure.

## Figures and Tables

**Figure 1 cancers-17-03679-f001:**
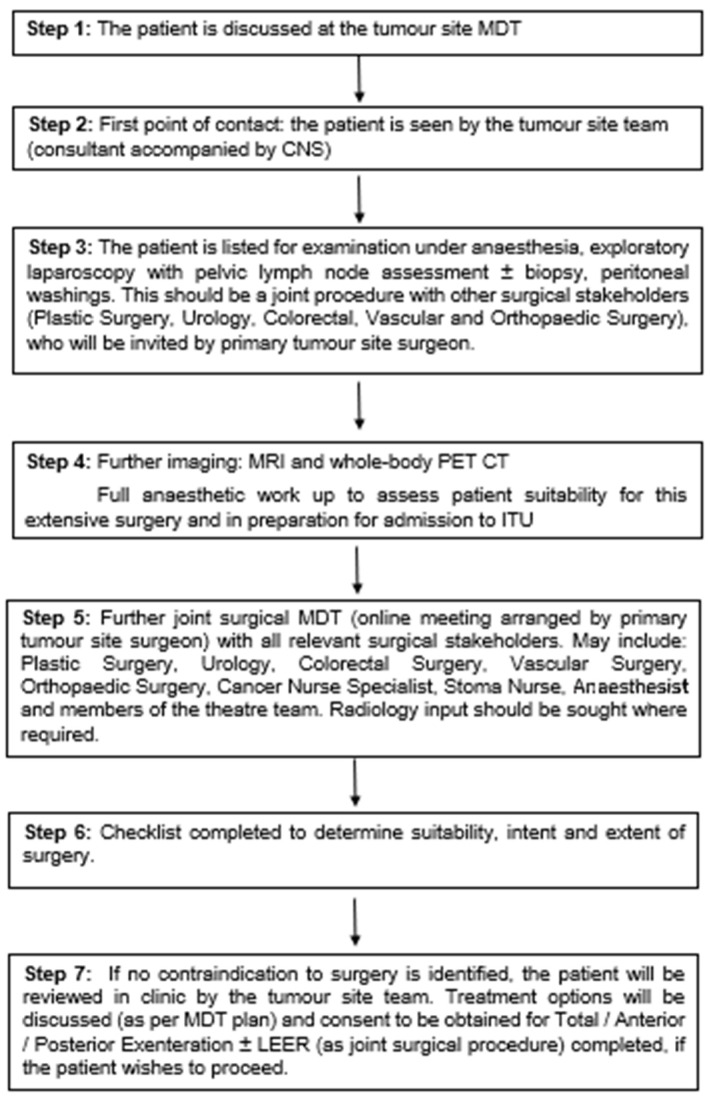
The Oxford pathway for patients undergoing pelvic exenteration ± laterally-extended endopelvic resection (LEER); MDT—multidisciplinary team meeting; CNS—cancer specialist nurses, ITU—Intensive Therapy Unit.

**Figure 2 cancers-17-03679-f002:**
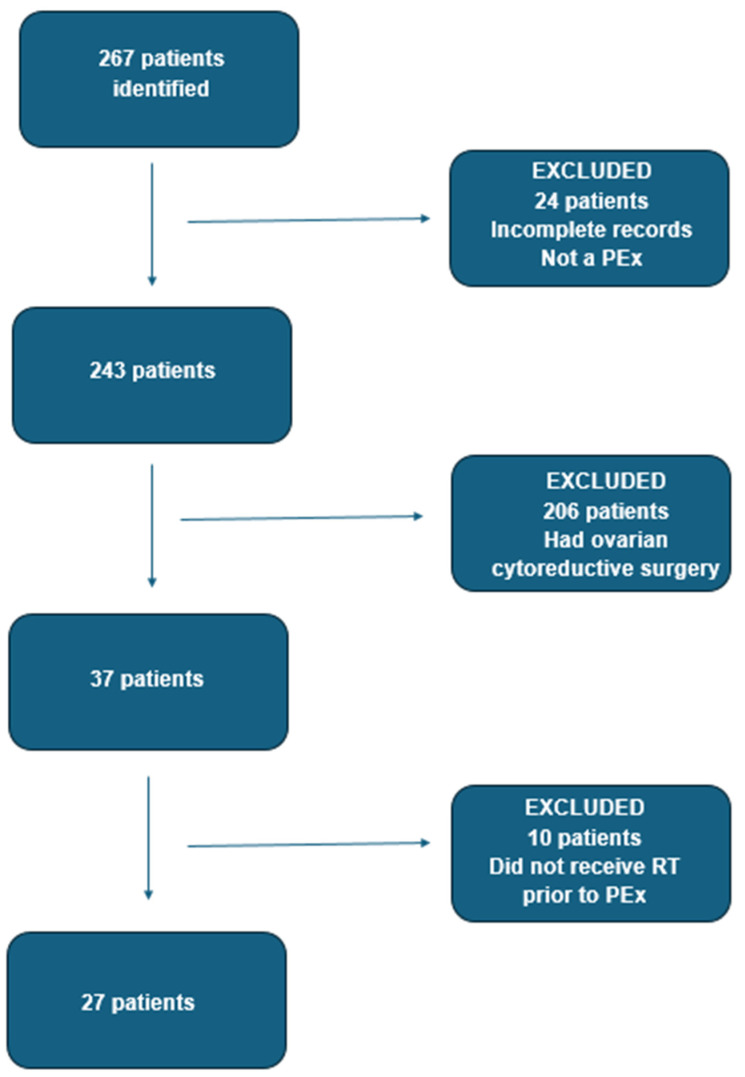
Flow-chart detailing patient selection (PEx—pelvic exenteration; RT—radiotherapy).

**Table 1 cancers-17-03679-t001:** Oxford checklist for pelvic exenteration and laterally extended endopelvic resections (LEER). MDT—multidisciplinary team meeting; CPEX—Cardiopulmonary exercise testing; MET—Metabolic Equivalent of Task.

**Patient/Main Surgeon Details**			
Name			
Date of birth			
Age			
Hospital number			
Primary tumour site surgeon			
Primary tumour site specialty			
**Investigations**			
Pathology of disease			
Last imaging date			
Relevant MDT outcome and date			
**Patient characteristics**			
Comorbidities			
Performance status			
CPEX studies (anaerobic threshold)			
ASA			
METs score			
**MDT**			
Relevant MDT outcome and date			
Seen by all relevant specialities			
Gynaecological Oncology	Yes	No	Not applicable
Colorectal	Yes	No	Not applicable
Urology	Yes	No	Not applicable
Plastics	Yes	No	Not applicable
Orthopaedics	Yes	No	Not applicable
Vascular	Yes	No	Not applicable
Anaesthetics	Yes	No	Not applicable
CNS	Yes	No	Not applicable
Stoma nurse	Yes	No	Not applicable
**Surgery planning**			
Operative intent: curative (R0) vs. palliative (R1)			
Proposed surgery			
Proposed date of surgery			
Are all relevant surgical teams available for proposed date of surgery?			
Is bowel preparation medication required and has it been prescribed?			
Has the patient has been fully counselled about impact of the surgery on their sexual function and quality of life?			
Is an Intensive Care/High Dependency Unit bed booked?			

**Table 2 cancers-17-03679-t002:** Patient characteristics and pre-operative assessment (BMI—body mass index; PS—performance status; ASA—American Society of Anaesthesiologists physical status classification system; CCI—Charlson Comorbidity Index; SCC—squamous cell carcinoma; CS—carcinosarcoma; G—grade; ADK—adenocarcinoma; EEC—endometrial endometrioid carcinoma; CR—chemoradiotherapy; S—surgery; C—chemotherapy; R—radiotherapy; FIGO—International Federation of Gynaecology and Obstetrics).

**Patient Characteristics**		
Age—median (range)	63	41–81
BMI—median (range)	27	17–45
PS		Number (%)
	0	18 (66.7%)
	1	8 (29.6%)
	2	1 (3.7%)
ASA		Number (%)
	1	18 (66.7%)
	2	7 (25.9%)
	3	2 (7.4%)
CCI—median (range)	2	0–5
CCI		Number (%)
	0	6 (22.2%)
	1	6 (22.2%)
	2	4 (14.8%)
	3	7 (25.9%)
	4	3 (11.1%)
	5	1 (3.7%)
**Primary tumour site/Histology**		
Vulva—number (%)		3 (11.1%)
	SCC G1	2
	SCC G2	1
Vagina—number (%)		4 (14.8%)
	SCC G2	1
	SCC G3	2
	ADK	1
Cervix—number (%)		11 (40.7%)
	SCC G3	9
	ADK	2
Uterus—number (%)		9 (33.3%)
	ADK	7
	CS	2
**FIGO stage at diagnosis**		
1	14 (51.9%)	
2	6 (22.2%)	
3	4 (14.8%)	
4	3 (11.1%)	
**Primary treatment type**		
	CR	12 (44.4%)
	CR + S	1 (3.7%)
	S+ CR	3 (11.1%)
	S + CR + C	1 (3.7%)
	S + R	8 (29.6%)
	S + R + C	2 (7.4%)
**Tumour pre-exenteration**		
	Recurrence	18 (66.7%)
	Persistent disease	6 (22.2%)
	New diagnosis	3 (11.1%)
Time to recurrence (months)—median (range)	33 (8–348)	
Tumour size in cm (imaging)—median (range)	4.1 (0.9–6.7)	

**Table 3 cancers-17-03679-t003:** Surgery (EBL—estimated blood loss; LOS—length of stay; APR—abdomino-perineal resection; VRAM—vertical rectus abdominis myocutaneous flap; ALT—antero-lateral thigh flap; SB—small bowel; LEER—laterally extended endopelvic resection).

Type of Surgery			
Exenteration type (anterior/posterior/total)—number (%)			
	Anterior	2 (7.4%)	
	Posterior	8 (29.6%)	
	Total	17 (63%)	
Exenteration type (I/II/III)—number (%)			
	I	15 (55.6%)	
	II	1 (3.7%)	
	III	11 (40.7%)	
LEER—number (%)			
	Yes	4 (14.8%)	
	No	23 (85.2%)	
Urinary diversion			
	No	8 (29.6%)	
	Yes	19 (70.4%)	
		Ileal Wallace	7 (25.9%)
		Ileal Bricker	3 (11.1%)
		Colonic Wallace	2 (7.4%)
		Colonic Bricker	7 (25.9%)
Colorectal surgery			
	No	2 (7.4%)	
	Yes	25 (92.6%)	
		Hartmann’s + end colostomy	16 (59.3%)
		APR + end colostomy	3 (11.1%)
		Hartmann’s + end colostomy + SB resection/anastomosis	1 (3.7%)
		Rectosigmoid resection (previous colostomy)	1 (3.7%)
		Excision of rectum/primary anastomosis (previous colostomy)	1 (3.7%)
		APR + end colostomy, SB resection/primary anastomosis	1 (3.7%)
		Resection of mid transverse colon to low rectum, SB resection and anastomosis (patient had prior ileostomy)	1 (3.7%)
		Rectosigmoid resection and primary anastomosis	1 (3.7%)
Plastic surgery			
	No	18 (66.7%)	
	Yes	9 (33.3%)	
		VRAM	6 (22.2%)
		ALT	2 (7.4%)
		Gracilis	1 (3.7%)
EBL	Median (range)		800 (150–2500)
LOS—days	Median (range)		18 (5–116)

**Table 4 cancers-17-03679-t004:** Complications (CD—Clavien–Dindo classification of postoperative complications within 30 days from surgery; VRAM—vertical rectus abdominis myocutaneous flap; EBL—estimated blood loss; PDS—polydioxanone; SB—small bowel; USS—ultrasound scan; HAP—hospital acquired pneumonia; AF—atrial fibrillation; SFA—superficial femoral artery; EUA—examination under anaesthesia).

**Intraoperative Complications**		
	No	19 (70.4%)
	Yes	8 (29.6%)
	Ischaemic VRAM flap (right inferior epigastric artery ligated); bilateral gracilis flaps sited instead	1 (3.7%)
	External iliac vein injury (sutured)—EBL 1.5 L	1 (3.7%)
	External. iliac vein injury (sutured)—EBL 2 L; Small ureteric laceration repaired with PDS suture	1 (3.7%)
	Cystotomy (repaired by Urology)	2 (7.4%)
	Iliac vein injury—EBL 2.5 L. Enterotomy (sutured).	1 (3.7%)
	Two SB enterotomies during adhesiolysis (multiple previous bowel surgeries)—patient underwent SB resection/primary anastomosis	1 (3.7%)
	Ischaemic Wallace ileal conduit—removed and Wallace colonic conduit formed instead	1 (3.7%)
**Postoperative complications (CD)**		
	1	9 (33.3%)
	2	12 (44.4%)
	3A	2 (7.4%)
	Paracolic collection (drained under USS guidance); hyponatraemia; HAP	
	Required left nephrostomy on single left kidney; AF post-op—resolved.	
	3B	4 (14.8%)
	VRAM flap failure—return to theatre day 2 postop. Perineal wound dehiscence. Pelvic collection. Urosepsis.	1 (3.7%)
	Entero-conduit and entero-vaginal fistulae—return to theatre day 9 postop for defunctioning loop ileostomy. Urine diverted by bilateral nephrostomies	1 (3.7%)
	Ischemic right leg—return to theatre day 1 postop for embolectomy (external iliac artery and SFA thrombus). Embolectomy failed. Recurrent ischemia secondary to compartment syndrome—return to theatre for femoro-femoral crossover graft + right lower leg fasciotomy.	1 (3.7%)
	Vaginal and anal stump dehiscence + rectovaginal fistula requiring EUA + washout + endosponge	1 (3.7%)

**Table 5 cancers-17-03679-t005:** Margin status. R0—clear histopathological margins, with tumour > 1 mm from margin; R1—microscopic residual disease < 1 mm from margin; R2—macroscopic residual disease at the time of surgery; SCC—squamous cell carcinoma; ADK—adenocarcinoma; CS—carcinosarcoma.

Histology/Margin Status		
Size (histology)—median (range)	4 (0.1–8.9)	
Margin status		
	R0	21 (77.8%)
	R1	6 (22.2%)
	R2	0
Number of margins involved for R1—median (range)		1 (1–3)
	1	4 (14.8%)
	2	1 (3.7%)
	3	1 (3.7%)
Pathology		
	SCC	11 (40.7%)
	ADK	10 (37%)
	CS	2 (7.4%)
	No residual tumour	4 (14.8%)
Location		
	Vulva	3 (11.1%)
	Vagina	19 (70.4%)
	Cervix	4 (14.8%)
	Uterus	3 (11.1%)
	Ovary	2 (7.4%)
	Bowel	10 (37%)
	Urethra	3 (11.1%)
	Bladder	6 (22.2%)
	Sidewall	3 (11.1%)
	Other	5 (18.5%)

**Table 6 cancers-17-03679-t006:** Statistical analysis—univariate analysis (N/A—no residual tumour; R0—clear histopathological margins, with tumour > 1 mm from margin; R1—microscopic residual disease < 1 mm from margin; SD—standard deviation; BMI—body mass index; PS—performance status; ASA—American Society of Anaesthesiologists score; CCI—Charlson comorbidity index; FIGO—International Federation of Gynaecology and Obstetrics; EBL—estimated blood loss; SCC—squamous cell carcinoma; ADK—adenocarcinoma; CS—carcinosarcoma).

	R0	R1	*p*	Test
Patients—number (%)	21 (77.7%)	6 (22.2%)		
Age—mean (SD)	59 (11.6)	70 (7)	0.015	Independent *t*-test
BMI—mean (SD)	29 (7.5)	29 (8.3)	0.495	Independent *t*-test
PS			0.243	Kruskal–Wallis
ASA			0.170	Kruskal–Wallis
CCI			0.065	Kruskal–Wallis
Primary tumour site			0.036	Fisher exact
Vulva	1	2		
Vagina	3	1		
Cervix	11	0		
Uterus	6	3		
Pathology			0.677	Fisher exact
SCC	8	3		
ADK	8	2		
CS	1	1		
No residual tumour	4	0		
Location (histopathology)				
Vulva	1	2	0.115	Fisher exact
Vagina	13	6	0.136	Fisher exact
Cervix	3	1	0.659	Fisher exact
Uterus	3	0	0.569	Fisher exact
Ovary	2	0	0.461	Fisher exact
Bowel	6	4	0.153	Fisher exact
Urethra	1	2	0.115	Fisher exact
Bladder	4	2	0.404	Fisher exact
Pelvic sidewall	2	1	0.545	Fisher exact
Other	3	2	0.303	Fisher exact
FIGO stage at diagnosis			0.066	Kruskal–Wallis
1	9	5		
2	5	1		
3	4	0		
4	3	0		
Tumour grade			0.333	Fisher exact
N/A	5	0		
1	1	2		
2	6	2		
3	9	2		
Tumour pre-exenteration			0.274	Fisher exact
Recurrence	12	6		
Persistent	6	0		
New	3	0		
Time to recurrence—mean (SD)	72(93)	34 (25)	0.170	Independent *t*-test
Tumour size (imaging)—mean (SD)	2.5 (1.9)	5 (1.2)	0.003	Independent *t*-test
Exenteration type			1	Fisher exact
Anterior	2	0		
Posterior	6	2		
Total	13	4		
Exenteration type			1	Fisher exact
I	12	3		
II	1	0		
III	8	3		
EBL—mean (SD)	643 (335)	1267 (794)	0.057	Independent *t*-test
Size(histology)—mean (SD)	2.6 (2.2)	6 (2)	0.001	Independent *t*-test

**Table 7 cancers-17-03679-t007:** Univariate logistic regression, with R0 as the dependent variable (BMI—body mass index; PS—performance status; ASA—American Society of Anaesthesiologists score; CCI—Charlson comorbidity index; FIGO—International Federation of Gynaecology and Obstetrics; LEER—laterally-extended endopelvic resection; EBL—estimated blood loss).

Dependent Variables	*p* Value	OR (95% CI)
Primary organ	0.278	0.167 (0.006–4.515)
Pathology	0.864	0.667 (0.087–5.127)
Age	0.054	1.127 (0.998–1.273)
PS	0.161	3.093 (0.637–15.028)
ASA	0.083	3.561 (0.848–14.955)
BMI	0.989	0.999 (0.883–1.131)
CCI	0.064	2.073 (0.960–4.478)
FIGO at diagnosis	0.129	0.228 (0.034–1.536)
Tumour grade	0.160	0.375 (0.095–1.475)
Time to recurrence	0.366	0.984 (0.950–1.019)
Tumour size (imaging)	0.038	2.905 (1.063–7.935)
Exent type ant/post/total	0.936	1.083 (0.154–7.642)
Exent type I/II/III	0.672	1.222 (0.483–3.094)
LEER	0.172	0.211 (0.022–1.972)
EBL	0.048	1.002 (1–1.005)
Tumour size (histology)	0.020	2.017 (1.119–3.635)
Histology vulva	0.086	10 (0.721–138.678)
Histology vagina	0.999	
Histology cervix	0.885	1.2 (0.101–14.195)
Histology uterus	0.999	
Histology ovary	0.999	
Histology bowel	0.105	5 (0.716–34.918)
Histology urethra	0.086	10 (0.721–138.678)
Histology bladder	0.464	2.125 (0.283–15,968)
Histology sidewall	0.628	1.9 (0.142–25.447)
Histology other	0.303	3 (0.370–27.295)

**Table 8 cancers-17-03679-t008:** Multivariable logistic regression, with R0 as the dependent variable (EBL—estimated blood loss).

Predictors	*p* Value	OR (95% CI)
Age	0.241	1.958 (0.939–4.085)
EBL	0.061	1.004 (1–1.008)
Tumour size (histology)	0.073	2.017 (1.119–3.635)

**Table 9 cancers-17-03679-t009:** Retrospective studies on pelvic exenteration for gynaecological malignancies published between 2015–2025. CD 3/4—Clavien–Dindo grade 3 and 4 complications.

N	Year	Author	Location	Period	No	Type of Cancer	Post RT Cases	R0	30-Days Severe Complications (CD3/4)	30-Days Mortality
1	2025	Nistor (current paper)	Oxford, UK	2011–2024	27	Cervical 11 (40.7%) Uterine 9 (33.3%) Vaginal 4 (14.4%) Vulval 3 (11.1%)	27 (100%)	21 (77.8%)	6 (23.2%)	0
2	2025	Plett [23]	Berlin, Germany	2016–2023	70	Cervical 48 (68.6%) Endometrial 8(11.4%) Ovarian 6 (8.6%) Vaginal 5 (7.1%) Vulvar 3 (4.3%)	36 (51.4%)	38 (54.2%)	27 (28.6%)	3 (4.3%)
3	2024	Tortorella [35]	Rome, Italy	2010–2019	129	Cervical 90 (69.8%) Endometrial 24 (18.6%) Vulval/vaginal 15 (11.6%)	106 (82.2%)	Not reported	36 (27.9%)	3 (2.3%)
4	2022	Rios-Doria [36]	New York, NY, USA	2010–2018	100	Cervical 30 (30%) Vulvar 27 (27%) Uterine 24 (24%) Vaginal 19 (19%)	Not reported	Not reported	50 (50%)	0
5	2022	Stanca [24]	Targu Mures, Romania	2010–2019	47	Cervical 47 (100%)	36 (76%)	30 (64%)	18 (38.3%)	3 (6%)
6	2021	Ter Glane [18]	Marburg, Germany	2011–2016	57	Cervical24 (51.1%) Vaginal 8 (17%) Vulval 5 (10.6%) Endometrial 4 (8.5%) Ovarian 2 (4.3%) Uterus 1 (2.1%) Others 3 (6.4%)	Not reported	30 (63.8%)	19 (40.4%)	2 (3.5%)
7	2021	Egger [37]	Bonn, Germany	2002–2016	49	Cervical 17 (34.7%) Vulvar 18 (36.7%) Endometrial 4 (8.2%) Anal cancer 1 (2%) Other 3 (6.1%) Vaginal 8 1(6.3%)	9 (18.3%)	35 (71.4%)	16 (32.7%)	1 (0.5%)
8	2021	Vigneswaran [38]	Chicago, IL, USA	2005–2016	335	Cervix: 105 (31.3) Ovarian: 91 (27.2) Uterine: 52 (15.5)	Not reported	Not reported	71 (21.2%)	4 (1.2%)
9	2021	Lewandowska [39]	Warsaw, Poland	2010–2018	44	Cervical 44 (100%)	43 (97.8%)	27 (61.4%)	11 (25%)	Not reported
10	2019	Matsuo [40]	Los Angeles, CA, USA	2001–2015	2647	Cervical 1194 (45.1%) Uterine 394 (14.9%) Vaginal 729 (27.6%) Vulval 328 (12.4%)	Not reported	Not reported	597 (22.6%)	49 (1.9%)
11	2019	Bacalbasa [41]	Bucharest, Romania	2014–2017	100	Cervical 58 (58%) Endometrial 13 (13%) Rectal 12 (12%) Ovarian 11 (11%) Vulvar 3 (3%) Vaginal 3 (3%)	48 (48%)	68 (68%)	22 (22%)	3 (3%)
12	2019	Kelly [2]	PelvEx Collaborative	2006–2017	523	Ovarian 224 (42.8%) Cervical 108 (20.6%) Endometrial 88 (16.8%) Vaginal 103 (19.7%)	Not reported	380(72.6%)	Ovarian 47 (20.9%) Cervical 37 (34.2%) Endometrial 23 (26.1%) Vaginal 49 (47.5%)	Ovarian 3 (1.3%) Cervical 2 (1.8%) Endometrial 2 (2.2%) Vaginal 1 (0.9%)
13	2018	Li [42]	Peking, China	2009–2016	38	Cervical 38 (100%)	27 (71%)	31 (81.6%)	21 (55%)	2 (5.3%) at 3 months
14	2017	Smith [28]	Columbus, OH, USA	2000–2015	144	Uterine 10 (7%) Cervical 38 (26%) Vulvar/vaginal 12 (8%) Ovarian 11 (8%) Bladder 30 (21%) Anal/Rectal 26 (18%) Colon 3 (2%) Multiple 4 (3%) Other 10 (7%)	66 (46%)	Not reported	106 (74%)	2 (1%)
15	2016	Graves [43]	Chicago, IL, USA	1999–2011	511	Cervical 511 (100%)	299 (58%)	Not reported	Not reported	15 (2.9%)

## Data Availability

The raw data supporting the conclusions of this article will be made available by the authors on request.

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
