# Peer review of "The Preoperative Prognosticators of Surgical Margins (R0 vs. R1) in Pelvic Exenteration—A 14-Year Retrospective Study from a Tertiary Referral Centre"

_cancers, 2025, doi:10.3390/cancers17223679_

Round 1

Reviewer 1 Report

Comments and Suggestions for Authors
  1. This is an interesting and important topic with less literature surrounding it than I expected. I think the authors should highlight the lack of previous literature in this space to allow the readeres to better appreciate their findings in the setting of a small sample size
  2. The use of only univariate analyses is troublesome. while your sample size is small it would be more scientifically sound to do multivariable logistic analysis or something similar to compare your significant variables to eachother and ensure they remain significant. 
  3. your paper has a few limitations but I do not see them highlighted in the discussion anywhere. please improve the scientific rigor of the paper by discussing relevant limitations of the study

Author Response

Comment 1: “This is an interesting and important topic with less literature surrounding it than I expected. I think the authors should highlight the lack of previous literature in this space to allow the readers to better appreciate their findings in the setting of a small sample size”

Response 1: Thank you for raising this issue. In response to this comment, we created table 9, comparing our findings with those of other retrospective studies published in the past 10 years, highlighting gaps in existing data.

The following paragraph has been added to the “Discussion” section. Recent literature has been summarised in Table 9.

“Our study is the first looking specifically at R0 rates in patients undergoing a pelvic exenteration for a non-ovarian gynaecological malignancy, following radiotherapy.  In table 9, we have summarised the findings of 15 retrospective studies  on pelvic exenteration published in the last 10 years (2015-2025), demonstrating data gaps, particularly in relation to reporting R0 rates (the main factor associated with survival) and post-radiotherapy status. “

Table  9 - Retrospective studies on pelvic exenteration for gynaecological malignancies published between 2015-2025. NSQIP - National Surgical Quality Improvement Program; NCDB – National Cancer Database

N

Year

Author

Location

Period

No

Type of cancer

Post RT cases

R0

Early severe complications

(CD3/4)

30-days mortality

1

2025

Nistor

Oxford, UK

2011-2024

27

Cervical 11 (40.7%)

Uterine 9 (33.3%)

Vaginal 4 (14.4%)

Vulval 3 (11.1%)

27 (100%)

21 (77.8%)

6 (23.2%)

0

2

2025

Plett

Berlin,

Germany

2016-2023

70

Cervical 48 (68.6%)

Endometrial 8(11.4%)

Ovarian 6 (8.6%)

Vaginal 5 (7.1%)

Vulvar 3 (4.3%)

36

(51.4%)

38 (54.2%)

27 (28.6%)

3 (4.3%)

3

2024

Tortorella

Rome,

Italy

2010-2019

129

Cervical 90 (69.8%)

Endometrial 24 (18.6%)

Vulval/vaginal 15 (11.6%)

106 (82.2%)

Not reported

36 (27.9%)

3 (2.3%)

4

2022

Rios-Doria

New York, USA

2010-2018

100

Cervical 30 (30%)

Vulvar 27 (27%)

Uterine 24 (24%)

Vaginal 19 (19%)

Not reported

Not reported

50 (50%)

0

5

2022

Stanca

Targu Mures,

Romania

2010-2019

47

Cervical 47 (100%)

36 (76%)

30 (64%)

18 (38.3%)

3 (6%)

6

2021

Ter Glane

Marburg, Germany

2011-2016

57

Cervical24(51.1%)

Vaginal 8 (17%)

Vulval 5 (10.6%) Endometrial 4 (8.5%)

Ovarian 2 (4.3%)

Uterus 1 (2.1%)

Others 3 (6.4%)

Not reported

30 (63.8%)

19(40.4%)

2 (3.5%)

7

2021

Egger

Bonn,

Germany

2002-2016

49

Cervical  17 (34.7%)

Vulvar 18 (36.7%)

Endometrial  4 (8.2%)

Anal cancer 1 (2%)

Other 3 (6.1%)

Vaginal  8 1(6.3%)

9 (18.3%)

35 (71.4%)

16 (32.7%)

1 (0.5%)

8

2021

Vigneswaran

NSQIP, USA

2005- 2016

335

Cervix: 105 (31.3)

Ovarian: 91 (27.2)

Uterine: 52 (15.5)

Not reported

Not reported

71(21.2%)

4 (1.2%)

9

2021

Lewandowska

Warsaw,

Poland

2010- 2018

44

Cervical 44 (100%)

43 (97.8%)

27 (61.4%)

11 (25%)

Not reported

10

2019

Matsuo

Nationwide Inpatient Sample,

USA

2001-2015

2647

Cervical 1194 (45.1%)

Uterine 394 (14.9%)

Vaginal 729 (27.6%)

Vulval 328 (12.4%)

Not reported

Not reported

597 (22.6%)

49 (1.9%)

11

2019

Bacalbasa

Bucharest,

Romania

2014-2017

100

Cervical  58 (58%)

Endometrial 13 (13%)

Rectal 12 (12%)

Ovarian 11 (11%)

Vulvar 3 (3%)

Vaginal 3 (3%)

48 (48%)

68 (68%)

22 (22%)

3 (3%)

12

2019

Kelly

PelvEx

Collaborative

2006 - 2017

523

Ovarian 224 (42.8%)

Cervical 108 (20.6%)

Endometrial 88 (16.8%)

Vaginal 103 (19.7%)

Not reported

380(72.6%)

Ovarian 47 (20.9%)

Cervical 37 (34.2%)

Endometrial 23 (26.1%)

Vaginal 49 (47.5%)

Ovarian 3 (1.3%)

Cervical 2 (1.8%)

Endometrial 2 (2.2%)

Vaginal 1 (0.9%)

13

2018

Li

Peking, China

2009-2016

38

Cervical 38 (100%)

27 (71%)

31 (81.6%)

21(55%)

2 (5.3%) at 3 months

14

2017

Smith

Ohio, USA

2000- 2015

144

Uterine 10 (7%)

Cervical 38 (26%)

Vulvar/vaginal 12 (8%)

Ovarian 11 (8%)

Bladder 30 (21%)

Anal/Rectal 26 (18%)

Colon 3 (2%)

Multiple 4 (3%)

Other 10 (7%)

66 (46%)

Not reported

106 (74%)

2 (1%)

15

2016

Graves

NCDB, USA

1999-2011

511

Cervical 511 (100%)

299 (58%)

Not reported

Not reported

15 (2.9%)

Comment 2: “The use of only univariate analyses is troublesome. while your sample size is small it would be more scientifically sound to do multivariable logistic analysis or something similar to compare your significant variables to eachother and ensure they remain significant.”

Response 2:  We have managed to perform multivariable analysis, using the following three variables: tumour size on histology, age, EBL. We did not include both tumour size on histology and tumour size on imaging as these two variables are strongly correlated.

Changes to text: The following table has been added at the end of the “ Results section”

Table 8. Univariate logistic regression, with R0 as the dependent variable (EBL – estimated blood loss)

Predictors

P value

OR (95% CI)

Age

0.241

1.958 (0.939 – 4.085)

EBL

0.061

1.004 (1 – 1.008)

Tumour size (histology)

0.073

2.017 (1.119 – 3.635)

We have also added the following sentence to the abstract: “. None of these variables retained significance in multivariable logistic regression.”

Comment 3: “Your paper has a few limitations but I do not see them highlighted in the discussion anywhere. please improve the scientific rigor of the paper by discussing relevant limitations of the study”

Response 3: We have acknowledged our study’s limitations in detail in the “Discussion” section

Changes to text:

This study has several important limitations that should be acknowledged. Firstly, its retrospective observational design is inherently subject to selection and information bias, as data were collected from existing medical records with variable completeness and accuracy. The small sample size (n = 27) limits the statistical power of the analyses and likely contributed to the fact that, although multivariable analysis was performed, none of the variables retained statistical significance. This limits the ability to draw firm conclusions regarding independent predictors of R0 resection. Pelvic exenteration is a rare procedure, reserved for a very limited number of patients, which inevitably results in a paucity of data. We believe it was important to restrict inclusion criteria to post-radiotherapy cases only, as achieving R0 resection in a previously irradiated field poses specific challenges. In gynaecological malignancies, radiotherapy typically delivers higher biologically effective doses (often exceeding 70–80 Gy EQD2 when combining external beam and brachytherapy) to central pelvic tissues, resulting in extensive fibrosis and obliteration of normal tissue planes. In contrast, colorectal and urological cancers usually receive lower or more conformally delivered doses—approximately 45–50 Gy for rectal cancer and 64–80 Gy for prostate or bladder cancer—targeting smaller volumes and sparing much of the surrounding soft tissue. Consequently, surgery after gynaecological radiotherapy is uniquely demanding, with increased risks of vascular and ureteric injury and difficulty distinguishing tumour from post-treatment fibrosis. Furthermore, we excluded ovarian cases, as en-bloc resections performed during ovarian cytoreductive surgery are typically supralevator and undertaken in non-irradiated fields, representing a distinct disease biology, treatment paradigm, and survival outlook. Our sample size is therefore comparable to other single-centre publications from major referral centres when accounting for these stricter inclusion criteria (Table 9). We believe that a future prospective, multicentre collaboration would be highly valuable to validate and build upon our findings. Finally, we do not report long-term oncological data as our cohort has not completed follow-up.

Reviewer 2 Report

Comments and Suggestions for Authors

Pelvic exenteration is often the only available potentially curative/radical treatment for patients with pelvic advanced or reccurent malignancies. Pelvic exenteration, when planned, need the risk estimation of possible and probable complications and final results. So the  paper is interresting for practical purposes. Resection R0 should be the most important purpose, as it is the most strong correlation factor with the succesful radical treatment. Prediction of such a result is important. Real prediction factors in analysed group of patients were size of malinancies, age of patients, location of the tumor and estimated blood loss, but the tumor size seemed to be the most important ones marker.  In the pelvis after radical radiotherapy it is very difficult to measure the real size of tumor, so the Authors propose to resect tumor with the 3cm margin.  Gould et al (Int J Surg 2022) in a systematic review have analyzed current literature and found that R0 resection is the most important factor in further disease free and overall survival time. 

I propose to correct the information  seen in the row 206 that Volume data are presented in Table 1 - they are in Table 2. The Authors should probably add the the systematic review of Gold et al paper to the references. 

Author Response

Comment 1: "I propose to correct the information  seen in the row 206 that Volume data are presented in Table 1 - they are in Table 2. The Authors should probably add the the systematic review of Gold et al paper to the references. "

We thank the reviewer for this comment. We have rectified line 206 as advised and have included the relevant publication by Gould et al (2022) as reference No 18. 

Reviewer 3 Report

Comments and Suggestions for Authors

Dear authors,

The paper addresses an important clinical question regarding prognostic factors for surgical margin status in pelvic exenteration. The retrospective design is appropriate given the nature of the study question, and the topic is relevant to gynecologic oncology. However, the study has several limitations that need to be addressed:

  1. The most significant limitation is the very small sample size (n=27). This severely limits the statistical power of the study and the generalizability of the findings. With such a small sample, it's difficult to draw firm conclusions about the true impact of any of the variables studied!!! Multivariable logistic regression could not be performed as well…this is crucial! If possible, the authors should attempt to expand the sample size by including data from other institutions or extending the study period. A larger sample would significantly improve the statistical power and reliability of the findings.
  2. The single-center design further limits generalizability. Surgical techniques, patient selection criteria, and postoperative management protocols can vary significantly across institutions. The retrospective nature also introduces potential biases related to data collection, documentation, and patient selection.
  3. The statistical methods used (univariate analysis) are appropriate for an initial exploration of the data, but they do not account for potential confounding variables. The lack of multivariable analysis is a major weakness, especially given the small sample size!
  4. There seem to be gaps in the data presented in tables and text. For example, in Table 5, the number of patients who did not have a residual tumour is not clear whether they fall into the RO or R1 category. The authors should carefully review the data presented in tables and text to ensure consistency and completeness.
  5. The study does not report a power analysis to justify the sample size. This would have helped demonstrate if the sample size was adequate to detect clinically significant differences.
  6. Given that pelvic exenteration is performed for a range of locally advanced pelvic malignancies, it is important to acknowledge that these tumors have different biological behaviors and sensitivities to treatment.

The discussion section should more thoroughly address the limitations of the study and compare the findings with those of other studies in the literature. The clinical implications of the findings should also be discussed in more detail.

Furthermore, I suggest - the authors should emphasize any unique aspects of their study (e.g., specific surgical techniques, patient selection criteria…etc) and explain how these aspects may contribute to the existing literature.

The bibliography is extremely sparse, given that there are many recent articles on the topic. Moreover, of the 29 bibliographic references, only 2 are less than 5 years old, and only 5 are less than 10 years old in total. This is unacceptable for a manuscript aspiring to be published in a journal such as Cancers.

Author Response

Comment 1: “The most significant limitation is the very small sample size (n=27). This severely limits the statistical power of the study and the generalizability of the findings. With such a small sample, it's difficult to draw firm conclusions about the true impact of any of the variables studied!!! Multivariable logistic regression could not be performed as well…this is crucial! If possible, the authors should attempt to expand the sample size by including data from other institutions or extending the study period. A larger sample would significantly improve the statistical power and reliability of the findings.”

We are grateful to the reviewer for this thoughtful and constructive comment. We fully acknowledge that the small sample size represents a major limitation of our study and inevitably affects the statistical power and generalizability of the findings. Pelvic exenteration is, however, a rare and highly complex procedure, performed only in a limited number of carefully selected patients at tertiary referral centres. Our study period already spans 14 years (2011–2024), which reflects the realistic case volume for a single regional centre managing post-radiotherapy gynaecological malignancies.

We fully agree that expanding the sample size through multicentre collaboration could enhance statistical robustness. However, multicentre data collection in pelvic exenteration has historically been challenging, given the marked heterogeneity in patient selection, institutional protocols, surgical approaches, and chemo- and radiotherapy regimens, which can introduce significant confounding and limit the comparability of outcomes. Furthermore, radiotherapy techniques and dose protocols have evolved substantially over the past decade—with improvements in conformal planning, image guidance, and dose delivery—such that including much older data may not accurately reflect current clinical practice or treatment-related tissue changes.

For these reasons, we prioritised a homogeneous, single-centre cohort to ensure internal consistency in patient selection, operative approach, and chemoradiotherapy background. We have elaborated on this rationale in the revised manuscript and emphasised that future prospective, multicentre collaborations—with harmonised data collection—would be highly valuable to validate and expand upon our findings. We are very grateful for the reviewer’s insightful feedback, which has helped us strengthen this section of the discussion.

Comment 2: “The single-center design further limits generalizability. Surgical techniques, patient selection criteria, and postoperative management protocols can vary significantly across institutions. The retrospective nature also introduces potential biases related to data collection, documentation, and patient selection.”

We respect and acknowledge this very helpful comment by the esteemed reviewer. We fully agree that the single-centre and retrospective design represents an inherent limitation of our study and may affect the generalizability of the findings. In an ideal scenario, standardisation of surgical techniques, patient selection criteria, and postoperative management protocols across institutions would indeed improve the comparability of results and ultimately strengthen global outcomes. However, such standardisation has proven challenging for highly complex gynaecological oncology pelvic exenterations, where surgical decisions are often influenced by patient-specific anatomy, tumour biology, prior treatments, and institutional expertise.

Our team recognises these limitations and has sought to contribute by publishing detailed institutional data to raise awareness within the gynaecological oncology community and encourage participation in future collaborative, multicentre networks aimed at harmonising practice and outcomes reporting. Despite the limited scale, our study reinforces the established finding that tumour size is associated with achieving an R0 resection, which is a well-recognised independent predictor of survival. We have clarified these points in the revised manuscript and are deeply grateful to the reviewer for highlighting this important aspect.

Comment 3: The statistical methods used (univariate analysis) are appropriate for an initial exploration of the data, but they do not account for potential confounding variables. The lack of multivariable analysis is a major weakness, especially given the small sample size!

We thank the reviewer for this very helpful comment. We are pleased to confirm that, following the reviewer’s suggestion, we have now performed a multivariable logistic regression analysis including the three most relevant variables: tumour size on histology, age, and estimated blood loss (EBL). We deliberately excluded tumour size on imaging, as it was strongly correlated with histological tumour size, and including both would have introduced collinearity. The results of this analysis have been added to the revised manuscript (Table 8, end of the Results section):

Table 8. Univariate logistic regression, with R0 as the dependent variable (EBL – estimated blood loss)

Predictors

P value

OR (95% CI)

Age

0.241

1.958 (0.939 – 4.085)

EBL

0.061

1.004 (1 – 1.008)

Tumour size (histology)

0.073

2.017 (1.119 – 3.635)

While none of the variables retained statistical significance in the multivariable model, likely due to the limited sample size, we acknowledge this as an important limitation and have highlighted it in the revised discussion. Nevertheless, we believe that clinicians are duty-bound to openly and widely share their findings, including complications, associated with this complex major pelvic surgery, and studies such as ours remain valuable in informing practice in this rare and technically demanding field. Despite its modest cohort, this work has directly contributed to the development and implementation of a Standard Operating Procedure for pelvic exenteration at our institution, leading to measurable improvements in service delivery and patient care.

Comment 4: There seem to be gaps in the data presented in tables and text. For example, in Table 5, the number of patients who did not have a residual tumour is not clear whether they fall into the RO or R1 category. The authors should carefully review the data presented in tables and text to ensure consistency and completeness.

We thank the reviewer for identifying this issue. We confirm that 4 out of the 27 patients had no residual tumour on the final histology specimen, and these cases were classified as R0 resections. We have reviewed the manuscript thoroughly and updated Tables 5 and 6 to ensure this information is presented clearly and consistently. We are grateful to the reviewer for highlighting this point, which has helped us improve the clarity and accuracy of the data presentation.

Comment 5: The study does not report a power analysis to justify the sample size. This would have helped demonstrate if the sample size was adequate to detect clinically significant differences.

We thank the reviewer for this thoughtful and important comment regarding sample size justification. Owing to the rarity and complexity of pelvic exenteration for non-ovarian gynaecological malignancies following prior radiotherapy, the study cohort is inherently small. This procedure is performed only infrequently, even in high-volume tertiary referral centres, making a priori power calculation impractical and potentially misleading. The primary aim of this study was therefore to provide a comprehensive descriptive analysis of outcomes and predictors in this highly selected and clinically challenging patient population, rather than to test a predefined hypothesis requiring statistical power.

We fully acknowledge the limitations associated with the small sample size and have now included a statement in the Methods and Limitations sections to clarify this point. We have also emphasised that our findings should be interpreted as exploratory and hypothesis-generating, intended to inform future multicentre collaborative studies.

Comment 6: Given that pelvic exenteration is performed for a range of locally advanced pelvic malignancies, it is important to acknowledge that these tumors have different biological behaviors and sensitivities to treatment.

We thank the reviewer for this very thoughtful and important observation. We fully agree that cervical, uterine, vulval, and vaginal cancers demonstrate distinct biological behaviours and sensitivities to treatment, which can influence both response to primary therapy and outcomes following pelvic exenteration.

We have added the following paragraph to the “Discussion” section:

“The observed differences likely reflect the distinct biological behaviours and treatment sensitivities among gynaecological malignancies. Cervical cancer is generally both radio- and chemosensitive, particularly in squamous histology, and tends to recur centrally, making it more amenable to complete surgical clearance. In contrast, endometrial cancers are often less radiosensitive and more prone to distant dissemination, reducing the likelihood of durable local control. Vulval cancers, while predominantly squamous, comprise two biologically distinct subtypes: HPV-associated (p16-positive) tumours, which are typically more radiosensitive and occur in younger patients, and p53-mutated (HPV-independent) tumours, which are more common in older women, often arise in the setting of chronic dermatoses, and are less responsive to radiotherapy. Vaginal cancers, though rare and often managed similarly to cervical cancer, may exhibit variable responses to radiation due to anatomical proximity to radiosensitive organs and reduced vascular perfusion in some segments of the vaginal wall, both of which limit the radiation dose that can be safely delivered and affect tissue tolerance. These differences underscore the importance of tailoring exenterative surgery to tumour biology, histological subtype, and recurrence pattern when selecting candidates for this complex and high-morbidity procedure.”

We thank the reviewer for prompting this valuable clarification.

Comment 7: The discussion section should more thoroughly address the limitations of the study and compare the findings with those of other studies in the literature. The clinical implications of the findings should also be discussed in more detail.

We have discussed the study limitations in detail in the “Discussion” section. We have added table 9, comparing our findings (R0 rates, Clavien-Dindo grade 3 and 4 complications, mortality rates) with those of 15 other retrospective studies published between 2015-2025.  The following paragraph has been added to the “Conclusion” section: “From a clinical standpoint, our findings emphasise the importance of meticulous preoperative evaluation, multidisciplinary planning, and judicious patient selection. The patients most likely to benefit from pelvic exenteration in a post-radiotherapy field are younger individuals with smaller, centrally recurrent tumours—particularly of cervical origin—where complete (R0) resection is achievable. Intraoperatively, careful control of blood loss, achieved through early devascularisation of the pelvis, is essential to optimising surgical precision and outcomes.

In summary, preoperative tumour size, primary tumour site, and patient age should be key considerations in selecting suitable candidates for pelvic exenteration following radiotherapy, while minimising intraoperative blood loss remains critical to maximising the likelihood of achieving R0 resection—the single most important determinant of survival in this complex procedure.

Comment 8: Furthermore, I suggest - the authors should emphasize any unique aspects of their study (e.g., specific surgical techniques, patient selection criteria…etc) and explain how these aspects may contribute to the existing literature.

We have addressed this in greater detail under the “Discussion” section, as follows:

“Our study is the first looking specifically at R0 rates and describing complications encountered in patients undergoing a pelvic exenteration for a non-ovarian gynaecological malignancy, following radiotherapy.  In table 9, we have summarised the findings of 14 retrospective studies on pelvic exenteration published in the last 10 years (2015-2025), demonstrating data gaps, particularly in relation to reporting R0 rates - the main factor associated with survival -  and post-radiotherapy status. Our study compares favorably with existing literature in terms of an R0 resection rate of 77.8% (54.2-81.6%) and a 30-days severe complications rate (Clavien Dindo grade 3 and 4) of 23.2% (21.2%-74%).”

Comment 9: The bibliography is extremely sparse, given that there are many recent articles on the topic. Moreover, of the 29 bibliographic references, only 2 are less than 5 years old, and only 5 are less than 10 years old in total. This is unacceptable for a manuscript aspiring to be published in a journal such as Cancers.

We thank the reviewer for this very relevant point. We have extended the literature review to include the following 11 studies published in the past 5 years:

22

Plett H, Ramspott JP, Büdeyri I, Miranda A, Sehouli J, Sayasneh A, Muallem MZ. Pelvic exenteration: an ultimate option in advanced gynecological malignancies—a single center experience. Cancers (Basel). 2025;17(14):2327. doi:10.3390/cancers17142327

23

Stanca M, Căpîlna DM, Căpîlna ME. Long-term survival, prognostic factors, and quality of life of patients undergoing pelvic exenteration for cervical cancer. Cancers (Basel). 2022;14(9):2346. doi:10.3390/cancers14092346

30

Nougaret S, et al. MRI predictors of R0 resection and survival outcomes after pelvic exenteration for gynecologic malignancies. Eur Radiol. 2025;35:2681–2691. doi:10.1007/s00330-024-10940-z

43

Di Donato V, Kontopantelis E, De Angelis E, Arseni RM, Santangelo G, Cibula D, Angioli R, Plotti F, Muzii L, Vizzielli G, Tozzi R, Chiantera V, Caruso G, Giannini A, Scambia G, Abu-Rustum NR, Benedetti Panici P, Bogani G; Pelvic Exenteration Study Group. Evaluation of survival and mortality in pelvic exenteration for gynecologic malignancies: a systematic review, meta-analyses, and meta-regression study. Int J Gynecol Cancer. 2025 Jun;35(6):101829. doi:10.1016/j.ijgc.2025.101829.

20

Barton D.P., Heath O.M., Shahnawaz R., Sheng Q., Alan T., Pardeep K. Pelvic exenteration for central pelvic cancer. In: Hoffman M., Hull T.L., Bochner B.H., editors. Major Complications of Female Pelvic Surgery. Cham: Springer; 2025. p. 427-39. doi:10.1007/978-3-031-66772-5_38

44

Classen-von Spee S, Baransi S, Fix N, Rawert F, Luengas-Würzinger V, Lippert R, Bonin-Hennig M, Mallmann P, Lampe B. Pelvic exenteration for recurrent vulvar cancer: a retrospective study. Cancers (Basel). 2024;16(2):276. doi:10.3390/cancers16020276.

34

Tortorella L, Marco C, Loverro M, Conte C, Persichetti E, Bizzarri N, Costantini B, Santullo F, Foschi N, Gallotta V, Avesani G, Chiantera V, Ercoli A, Fanfani F, Fagotti A, Mele MC, Restaino S, Gueli Alletti S, Scambia G, Vizzielli G. Predictive factors of surgical complications after pelvic exenteration for gynecological malignancies: a large single-institution experience. J Gynecol Oncol. 2024;35(1):e4. doi:10.3802/jgo.2024.35.e4

35

Rios-Doria E, Filippova OT, Straubhar AM, Chi A, Awowole I, Sandhu J, Broach V, Mueller JJ, Gardner GJ, Jewell EL, Zivanovic O, Leitao MM Jr, Roche KL, Abu-Rustum NR, Sonoda Y. A modern-day experience with Brunschwig’s operation: outcomes associated with pelvic exenteration. Gynecol Oncol. 2022;167(2):277-282. doi:10.1016/j.ygyno.2022.08.017

36

Egger EK, Liesenfeld H, Stope MB, Recker F, Döser A, Könsgen D, Marinova M, Hilbert T, Exner D, Ellinger J, Mustea A. Pelvic exenteration in advanced gynecologic malignancies – who will benefit? Anticancer Res. 2021;41(6):3037-43. doi:10.21873/anticanres.15086

37

Vigneswaran HT, Schwarzman LS, Madueke IC, David SM, Nordenstam J, Moreira D, et al. Morbidity and mortality of total pelvic exenteration for malignancy in the US. Ann Surg Oncol. 2021;28(8):2790-2800. doi:10.1245/s10434-020-09247-2.

38

Lewandowska A, Szubert S, Koper K, Koper A, Cwynar G, Wicherek Ł. Analysis of long-term outcomes in 44 patients following pelvic exenteration due to cervical cancer. World J Surg Oncol. 2020;18:234. doi:10.1186/s12957-020-01997-3.

Reviewer 4 Report

Comments and Suggestions for Authors

This study demonstrated the preoperative risk factors for anR0/ R1 pelvic exenteration for gynaeclogical cancers. They demonstrate type of tumour, size, advanced age and blood loss as significant. The authors should explain why other important risk factors such as the biology (histological grade) of the tumour- differentiation, the presence of lateral pelvic lymph nodes and micro/ macrometastatic disease (M1) were not considered in order to give more validity of the study, Tumour size can be a risk factor surgically  but a small tumour may also have poor biology with high risk of local and distant recurrence. Preoperative chemoradiotherapy would not change the original biology of the tumour even it may downstage or downsize the tumour. Infact, other studies did not find tumour size a risk factor. These should be discussed in the limitations of the study.

Author Response

Comment 1: This study demonstrated the preoperative risk factors for anR0/ R1 pelvic exenteration for gynaeclogical cancers. They demonstrate type of tumour, size, advanced age and blood loss as significant. The authors should explain why other important risk factors such as the biology (histological grade) of the tumour- differentiation, the presence of lateral pelvic lymph nodes and micro/ macrometastatic disease (M1) were not considered in order to give more validity of the study, Tumour size can be a risk factor surgically  but a small tumour may also have poor biology with high risk of local and distant recurrence. Preoperative chemoradiotherapy would not change the original biology of the tumour even it may downstage or downsize the tumour. Infact, other studies did not find tumour size a risk factor. These should be discussed in the limitations of the study.

We thank the reviewers for raising this issue.  We have added our data on tumour grade to the results section in Table 6. We did not identify a statistically-significant association between tumour grade ad R0 rates. We have also performed univariable logistics regression but no association was identified. We have updated table 7:

Tumour grade

0.160

0.375 (0.095 – 1.475)

We have added the following paragraph under “Results”:

3.4 Lymph node status

With regards to lymph node status, none of our patients had macroscopically-involved lymph nodes. Microscopic lymph node involvement was identified in two cases. The first case, a 74yo patient undergoing pelvic exenteration for uterine carcino-sarcoma, with a history of anal squamous cell carcinoma treated with chemoradiotherapy, was found to have microscopic involvement of para-aortic lymph nodes by metastatic anal squamous cell carcinoma. This patient had clear margins (R0). The second patient, a 63yo undergoing posterior exenteration for recurrent vulval squamous cell carcinoma, was found to have a microscopically-involved left femoral lymph node. This patient had a tumour measuring 8.9cm on histology and the margins were microscopically involved (R1). We did not include the lymph node status in our statistical analysis as we only had two cases with lymph node involvement, one by a different tumour.

The following paragraph has been added under “Limitations”:

“Positive pelvic, para-aortic (43) and inguino-femoral (44) nodal involvement is known to be a key predictor of reduced overall survival. Our study only identified one patient with an unexpected microscopic involvement of para-aortic lymph nodes by a different 

Round 2

Reviewer 3 Report

Comments and Suggestions for Authors

Congratulations! The authors respected all the indications mentioned in the first review!